# A Case-Series Report on The Use of a Salicylic Acid Bandage as a Non-Antibiotic Treatment for Early Detected, Non-Complicated Interdigital Phlegmon in Dairy Cows

**DOI:** 10.3390/ani9040129

**Published:** 2019-03-29

**Authors:** Ylva Persson, Marie Jansson Mörk, Märit Pringle, Christer Bergsten

**Affiliations:** 1Department of Animal Health and Antibiotic Strategies, National Veterinary Institute, 751 89 Uppsala, Sweden; marit.pringle@sva.se; 2Växa Sverige, Box 30204, 104 25 Stockholm, Sweden; mariejanssonmork@gmail.com; 3Department of Biosystems and Technology, Swedish University of Agricultural Sciences, P.O. Box 103, 230 53 Alnarp, Sweden; christer.bergsten@slu.se

**Keywords:** lameness, foot rot, *Fusobacterium necrophorum*

## Abstract

**Simple Summary:**

Interdigital phlegmon (IP) is a common and economically important cause of acute lameness in cattle. It is most often treated with antibiotics. To reduce the use of antibiotics, we wanted to evaluate the effect of treatment with a salicylic acid bandage of the hoof of early-detected IP in dairy cows. We found that treatment by farmers of mild IP with salicylic acid was generally satisfactory. Within three to five days, treated cows responded with reduced lameness, lower body temperature, decreased swelling, and an improved general condition compared with the day when the treatment started. The salicylic acid bandage therefore showed the potential to be an attractive alternative in the treatment of early detected, non-complicated IP. The benefits of using salicylic acid instead of antibiotics include reduced risk of antimicrobial resistance, no painful injections of antibiotics, cheaper treatment costs, and no withdrawal of milk. However, it is important that the diagnosis is correct and that other claw disorders and complications are identified and treated promptly.

**Abstract:**

Interdigital phlegmon (IP) is an important cause of lameness in cattle. The aim of this study was to evaluate the treatment effect of a salicylic acid bandage in the interdigital space in dairy cows with early detected IP. Dairy cows (n = 109) with IP diagnosed and treated by the farmer were included in the study. On day 0, the rectal temperature, general condition, coronary circumference, and lameness score were recorded. The cow was immobilized in a trimming chute and the interdigital space was cleaned and inspected. For treatment, 1–2 tablespoons of 100% salicylic acid powder were applied into the interdigital space followed by bandaging of the hoof. On days 1–2 and days 3–5, the rectal temperature, the general condition, and the lameness score were recorded. On days 3–5, the cow was restrained, the bandage was taken off, and the coronary circumference was recorded again. Treatment of IP with salicylic acid gave a satisfactory treatment result. Within three–five days, treated cows responded with reduced lameness, lower body temperature, decreased coronary swelling, and an improved general condition compared to the day when the treatment started. Salicylic acid therefore proved to be an alternative in the treatment of early-detected non-complicated IP.

## 1. Introduction

Interdigital phlegmon (IP, foot rot, foul-in-the-foot) is a common and economically-important [1] infectious cause of acute lameness in dairy cattle, especially in loose-housed cattle, or cattle on pasture. Outbreaks are often seen when animals are moved or traded, e.g., when new herds are established. Also, beef cattle and fattening bulls can be affected; up to 36% of fattening bull calves were treated in a Swedish herd outbreak [2]. The veterinary treatment incidence for IP in Sweden is <1 per 100 cows per year (Swedish Board of Agriculture 2017, Marie Mörk, Växa Sverige 2017). However, not all cases of IP are reported, so the true incidence is probably slightly higher [3].

The key pathogen in the infection is *Fusobacterium (F.) necrophorum,* an obligate anaerobic bacterium. Many other bacteria are often present in the lesion, but it is not clear whether they are pathogenic or part of the normal microbiota in the interdigital space [4]. *Fusobacterium necrophorum* is susceptible to penicillin and there are no signs of emerging antibiotic resistance according to a Swedish surveillance of cows with clinical IP, where all isolates had a MIC (minimal inhibitory concentration) of penicillin ≤0.06 mg/L [5]. Similar resistance patterns for *F. necrophorum* have been reported from other parts of the world [6,7].

According to Swedish guidelines for antibiotic use in cattle (antibiotics can only be prescribed by a veterinarian), the first drug of choice for the treatment of acute clinical IP is systemic treatment with benzylpenicillin for three days [8]. Penicillin treatment, which is used in 85% of cases, is effective according to Swedish studies [9,10] and reports from the field. Moreover, in recommendations from the Swedish Medical Products Agency, prolonged action benzylpenicillin can be used to treat bovine IP with a single dose injection [11]. Based on resistance data, there are no justifications for using broad-spectrum antibiotics, like tetracycline, used in 15% of cases in Sweden, when treating foot rot in cattle. The highly prioritized critically important (for humans) antibiotic ceftiofur should be avoided (use is restricted in Sweden) due to the risk of emerging antimicrobial resistance (AMR). Although benzylpenicillin is a narrow spectrum antibiotic with less impact on the selection and spread of AMR, a treatment alternative without antibiotics would be even more beneficial from an AMR point of view. According to Haggman et al. 2015 [1], outbreaks of IP cause economic losses to the dairy industry, and the majority of the costs come from the discarded milk due to the treatments with antibiotics, so a non-antibiotic alternative treatment with no withdrawal time would be desirable.

There are many reports from the field in Sweden that IP is successfully treated with local salicylic acid and a bandage. In earlier studies, salicylic acid has been successfully used for the treatment of digital dermatitis [12,13,14,15]. However, the effect of salicylic acid as a treatment of IP has not been evaluated. 

Salicylic acid is a substance with keratolytic (or desmolytic) properties [16]. This mechanism of action causes salicylic acid to be found in some skin care products, including medical ones. Salicylic acid is also a non-steroid anti-inflammatory drug (NSAID) and has some anti-inflammatory properties [16]. Salicylic acid may also have some antibacterial properties, via its inhibition of bacterial communication (*quorum sensing*), as shown for *Pseudomonas aeruginosa* [17]. This causes loss of some bacterial virulence properties, such as the ability to form toxins and biofilms. Today, salicylic acid is used primarily for external use because it may cause side effects as it irritates the intestines [18]. 

The aim of this case-series report was to evaluate the effect of local administration of salicylic acid in the interdigital space protected with a bandage in dairy cows with early detected, non-complicated IP. 

## 2. Materials and Methods 

Dairy cows with non-complicated IP and no other signs of disease were included in the study. The farmer was responsible for diagnosis, sampling, treatment, follow-up, and record keeping. The farmers participated voluntarily and had earlier experience of either the treatment method and/or treatment of lame cows with other claw disorders and had a trimming chute. All farmers received a package with salicylic acid, an elastic bandage (Kruuse Vet-Flex), and instructions of clinical investigation and bacteriological sampling. The clinical instructions included a photo atlas with a description of lameness scoring (https://www.zinpro.com/lameness/dairy/locomotion-scoring) adapted (0–4 instead of 1–5) and translated (Swedish) from Sprecher [19], a photo atlas of claw disorders (https://www.vxa.se/globalassets/dokument/fordjupningar/se-claw-atlas-2013-08-29-webb.pdf), and a step by step instruction on how to perform the treatment. On day 0, the rectal temperature, general condition, and lameness score were recorded. The cow was immobilized in a trimming chute, and the interdigital space was cleaned with soap and water and inspected. The coronary circumference was recorded (mm) with a soft measuring tape. From the first 5 cows in each herd, a sample for bacteriological culture was taken with an Amie´s culturette from the cleaned interdigital space and sent to the National Veterinary Institute (SVA) the same day. For treatment, 1–2 tablespoons of 100% salicylic acid powder (Kruuse) were administered into the interdigital space followed by bandaging the hoof. On days 1–2, the rectal temperature, the general condition, and the lameness score were recorded. General condition was scored as affected (1) or not (0). Lameness was scored as unaffected (0), mildly lame (1), moderately lame (2), lame (3) or severely lame (4). On days 3–5, the cow was again restrained in a trimming chute, and the bandage was taken off. The rectal temperature, general condition, and lameness score were again recorded, as was the coronary circumference (mm). The farmers were also asked to give their opinion regarding the treatment procedure. The clinical protocols were then sent to SVA. 

The bacteriological samples were cultured on fastidious anaerobe agar with 10% horse blood (National Veterinary Institute, Sweden) in an anaerobic atmosphere for two days at 37 °C. Colonies with the typical appearance of *Fusobacterium* spp. were analyzed by matrix-assisted-laser desorption/ionization time-of-flight mass spectrometry (MALDI-TOF MS).

The antimicrobial susceptibility tests were performed by broth microdilution (VetMIC, National Veterinary Institute, Sweden) in cation-adjusted Mueller Hinton broth. The inoculum was prepared by colony suspension to a concentration of approximately 10^6^ CFU/ml. Each well in the microtiter plate was inoculated with 100 μl. The MIC was read as the lowest antimicrobial concentration completely inhibiting visible growth. *Fusobacterium necrophorum* subsp. *necrophorum* (CCUG 9994) was used as a control strain.

The development from day 0 to days 1–2 and days 3–5 for temperature, general condition, and lameness was documented. Swelling was only measured on day 0 and days 3–5 when the bandage was removed. An improved temperature was identified as a decrease from >39.1 °C to ≤39.1 °C. Improved swelling was identified as a reduction in coronary circumference by ≥10 mm. For general condition and lameness, the proportions of cows for which the symptom improved, deteriorated, or remained unchanged during the follow-up period were identified. 

Differences in temperature between day 0 and days 1–2 and between day 0 and days 3–5 were analyzed using a paired t-test. Differences in swelling (coronary circumference) between day 0 and days 3–5 were analyzed using the Kolmogorov-Smirnov equality-of-distributions test (due to the data not being normally distributed). Prevalence and the 95% confidence intervals of lame cows and cows with the affected general condition were estimated for days 1–2, days 3–5, and for day 0. Non-overlapping confidence intervals were counted as a statistically significant difference, only including the cows in day 0 that also had a follow-up at days 1–2 and 3–5, respectively. Cows treated with antimicrobials were excluded from the analyses. 

Originally, it was planned to have a positive control group with systemic treatment with benzylpenicillin. However, it turned out to be very difficult to recruit veterinary practitioners willing to participate, so this treatment group was not included. 

A long-term follow-up was made at 6 and at 12 months from the initial treatment based on data on culling and veterinary treatments retrieved from the cow recording scheme database. 

## 3. Results

In total, 109 cases of acute clinical interdigital phlegmon from 24 herds entered the study. From each herd, between 1–35 cases (median 3 cases) were submitted to SVA. For 2 of the cases, only culturettes for culturing, and no records, were submitted. Seven cows were initially treated with both a local salicylic acid bandage and antibiotics and were therefore excluded from the study. These cows had an initial mean body temperature of 39.3 °C (day 0), followed by a mean body temperature of 39.2 °C (days 1–2) and 39.3 °C (days 3–5). Mean days in milk (DIM) for cows treated only with salicylic acid (n = 100) was 83 days, and 46 days for cows treated also with antibiotics (n = 7). 

Samples from 60 cows were cultured, and of these 30 were positive for *F. necrophorum*. The results of the antimicrobial susceptibility testing of 24 of these isolates are presented in Table 1. All isolates were susceptible to penicillin.

For 88 of the remaining 100 cows (those with a record and no antibiotic treatment), the infection was located to one of the hind feet. Temperature and coronary circumference at day 0 and at the follow-up are presented in Table 2 for all cows with the symptom reported on day 0. Similarly, general condition and lameness scoring at day 0 and at the follow-up are presented in Table 3.

Of the 90 cows, a temperature reported on day 0 and days 1–2, 61.1% had a fever (≥39.1 °C) on day 0 and 22.2% had a fever on days 1–2. Similarly, of the 82 cows with reported temperature on day 0 and days 3–5, 58.5% and 13.4%, respectively, had a fever. 

For 48 of the 63 cows (75.4%) for which the circumference was measured at both day 0 and days 3–5, the circumference was reduced with ≥10 mm. Both temperature and circumference were lower at follow-up compared to day 0 (Table 4). 

Of the 81 cows with the reported general condition at both day 0 and days 3–5, 24 were improved at days 3–5, 48 remained unaffected, 8 remained affected, and 1 was impaired. Of the 83 with lameness scoring on both days 0 and 3–5, 20 cows remained not lame at days 3–5, 56 cows had improved, and 26 remained lame. Both the proportion of cows with affected general condition and the proportion of lame cows were lower at days 3–5 compared to day 0, but there was no statistically significant difference between day 0 and days 1–2 (Table 5).

Seven of the 100 included cows were culled within six months after treatment with salicylic acid, of which one was culled because of a leg disorder. Within a year after treatment, 20 of the 100 cows were culled, of which four were due to a foot and leg disorder (reduced fertility was the most common culling reason). Of the seven cows treated with antibiotics, one was sent to slaughter within 12 months due to poor udder health. 

Two of the 100 cows had a second IP treatment recorded at 27 and 36 days, one at 284 days after the initial treatment with salicylic acid, and one cow had a third IP recorded 3 weeks after the previous treatment. 

## 4. Discussion

For dairy cows with early detected interdigital phlegmon, treatment with a salicylic acid bandage resulted in reduced body temperature, improved general condition, less lameness, and reduced coronary circumference within five days of treatment. However, the initial fever was lower than normally seen in acute interdigital phlegmon [9], indicating that only less severe cases were included in this study. Bergsten and Carlsson [2] saw the same reduction in body temperature in finishing cattle treated for IP (0,6 °C–0,9 °C) but the initial fever was higher than in this study (40.2 °C vs. 39.2 °C). The coronary circumference reflects the local swelling and the reduction seen in this study was as reported by Bergsten and Carlsson [2]. 

A decrease in lameness score is the most obvious symptom of an effective treatment while fever and swelling can persist for longer, as was observed in an antibiotic treatment study [2]. In our study, a decrease in lameness score was not seen until days 3–5, which might have resulted in some of the farmers and the veterinarians considering the use of antibiotics. In 7 of the 107 cows, the farmer and the practitioner chose to treat the cow with systemic penicillin rather quickly after day 0, probably due to a more severe inflammation, where salicylic acid was thought not to be enough. Despite antibiotic treatment, these cows had no decrease in mean body temperature at days 3–5 compared to day 0. It is thus always important, irrespective of treatment, to follow up the progress of the disease and be aware of the differential diagnosis and to reconsider treatment if lameness persists. However, the procedure with the salicylic acid bandage requires that the cow and the foot are immobilized, which makes it easier to confirm the presence of the IP lesion in the interdigital skin. Or, alternatively, to find another lesion causing the lameness. In contrast, when systemic antibiotics are used, there is commonly no thorough investigation of the foot. Some farmers found the bandaging procedure time-consuming, and one farmer complained about lesions in the coronary skin from improper bandaging. Thus, it is important that the bandaging technique is properly described and taught. Nevertheless, even with the extra work, many farmers in this study were satisfied with the treatment with salicylic acid and stated that the treatment was efficient. None of the cows treated either with the salicylic bandage alone or with antibiotics seemed to suffer from an immediate relapse within about a month after the initial treatment and culling due to foot and leg disorders did not differ from the Swedish average. 

Published data regarding antibiotic susceptibility of *F. necrophorum* in cattle is scarce and no established breakpoints for either microbiological or clinical resistance are available. For the beta-lactam antibiotics (penicillin and cephalothin) and tetracycline, the MICs were low. It is likely that the isolates had no acquired resistance to other tested substances but the MICs for fluoroquinolones, macrolides, and trimethoprim were higher or at the limit of what is commonly considered to be the clinically susceptible category for other bacteria. These results are in accordance with a previous Swedish study [5]. From a resistance perspective, it is both realistic and desirable that no IP cases should be treated with tetracycline or other broad-spectrum antibiotics, especially not with newer generations of cephalosporines, since they are considered among the most highly critically important antibiotics for humans.

To our knowledge, this is the first study on the treatment effect of salicylic acid in early detected, non-complicated IP. Schultz and Capion [12] have earlier demonstrated a good treatment efficacy for treatment of digital dermatitis with a salicylic bandage compared to tetracycline spray. 

One major weakness of this study is the lack of a control group. Due to welfare issues, it was not possible to include a negative control group. A positive control group treated with penicillin was not included because it proved difficult to recruit field veterinarians (required for antibiotic and NSAID treatment in Sweden). Another weakness is that the diagnosis of IP was made by farmers and not by veterinary practitioners. However, because the treatment required immobilization of the cow and foot in a trimming chute, cleaning of the interdigital skin and in many cases a bacterial swab in the wounded skin, it is likely that there was a thorough investigation of the interdigital skin in most cases, increasing the likelihood of a correct diagnosis. Moreover, Swedish farmers, not least in organic farms where milk withdrawal times are doubled, are motivated to treat IP without antibiotics, and salicylic acid is already used in many dairy farms to treat IP without the involvement of a veterinarian.

Thus, despite these weaknesses, we believe that this paper provides good evidence that a bandage with salicylic acid may be a useful alternative to antibiotics in the treatment of non-complicated cases of IP in dairy cows, provided that the herdsperson is alert to early symptoms. However, a randomized positive-controlled, blinded study is necessary to provide evidence that would allow us to firmly recommend this treatment instead of systemic antibiotics. In beef cattle or in young stock, this method of treatment is likely to be impractical and thus less relevant. Even more importantly, we do not claim or recommend that local treatment with salicylic acid should be used instead of systemic antibiotics treatment in later detected or more severe cases of IP. In severe cases there is always a risk for complications and failure in treatment despite the use of antibiotics [4]. Treatment with salicylic acid is a relatively cheap treatment that can be performed by the farmer or an educated herdsperson without withdrawal time for the milk. Additionally, in the Swedish system, an important benefit of salicylic acid treatment is that it can be initiated earlier than if a veterinarian is required for the antibiotic treatment. 

Treatment with salicylic acid could possibly be combined with NSAIDs; however, there are no published studies reporting the impact of NSAID treatment on IP, and in this study, we tested the effect of salicylic acid alone, as an additional NSAID treatment would have made the treatment outcome more difficult to interpret.

Three goals of successful treatment have all been fulfilled in this study: Cows were satisfactory cured and discharged milk and cost of treatment were reduced. The study also contributes to the overall goal of minimizing the use of antibiotics and the risk of resistance. 

## 5. Conclusions

Treatment with salicylic acid locally in the interdigital space of early-observed non-complicated interdigital phlegmon gave a satisfactory treatment result. Within five days, treated cows responded with reduced lameness, lower body temperature, decreased coronary swelling, and an improved general condition compared with the day when the treatment started. Salicylic acid is therefore an attractive non-antibiotic alternative in the treatment of non-complicated IP. Because it is inexpensive and easy to use and treatment can be performed by an educated herdsperson, it has no withdrawal time and it does not contribute to the antibiotic resistance problem. The cow must be handled in a trimming chute or correspondingly. However, our conclusions need to be tested in a randomized, positive-controlled, blinded study.

## Figures and Tables

**Table 1 animals-09-00129-t001:** Distribution (No. of isolates) of MICs of 6 antimicrobial agents for 24 isolates of *Fusobacterium necrophorum*.

Antimicrobial Agent	Distribution (No. of Isolates) of MIC ^a^ (mg/L)											
	≤0.016	0.03	0.06	0.12	0.25	0.5	1	2	4	8	16	>16
Cephalothin				6	18							
Ciprofloxacin						1	23					
Erythromycin									15	9		
Penicillin	1	18	5									
Tetracycline					21	3						
Trimethoprim										1	7	16

^a^ White fields denote the range of dilutions tested for each substance. MICs above the range are given as the concentration closest to the range. MICs equal to or lower than the lowest concentration tested are given as the lowest tested concentration. MIC, minimal inhibitory concentration.

**Table 2 animals-09-00129-t002:** Temperature (°C) and circumference (mm) at day 0 and at the follow-up for all cows with reported symptom at day 0.

Day	Temperature (°C)				Circumference (mm)			
	n	Mean	(Range)	Standard Deviation	n	Mean	(Range)	Standard Deviation
Day 0	98	39.2	(37.5–40.8)	0.55	68	303.4	(210–460)	63.0
Day 1–2 *	90	38.8	(37.6–40.4)	0.53	-			
Day 3–5	82	38.6	(37.4–39.9)	0.45	60	281.4	(200–420)	60.0

* Circumference not measured at day 1–2 because of bandage.

**Table 3 animals-09-00129-t003:** General condition and lameness scoring at day 0 and at the follow-up for all cows with reported symptom at day 0.

	General Condition		Lameness					
Day	n	Percent Affected	n	Score (Percent of Cows)				
				0	1	2	3	4
Day 0	97	38.1	98	1.0	27.6	53.1	17.4	1.0
Day 1–2	90	16.7	90	7.8	46.7	37.8	7.8	0.0
Day 3–5	82	11.0	84	39.3	52.4	6.0	2.4	0.0

**Table 4 animals-09-00129-t004:** Analysis of change in temperature (°C) and general condition from day 0 to days 1–2 and 3–5, using a paired t-test for temperature and the Kolmogorov-Smirnov equality-of-distributions test for the general condition.

Clinical Parameters	n	Mean (SE *)		*p*-Value
		Day 0	Follow-up	
Temperature day 0 compared to day 1–2	90	39.1 (0.06)	38.8 (0.06)	<0.001
Temperature day 0 compared to day 3–5	82	39.3 (0.05)	38.6 (0.05)	<0.001
Circumference day 0 compared to day 3–5	63	314.3	283.8	0.04

* Standard error, paired t-test.

**Table 5 animals-09-00129-t005:** The proportion of cows (%) and 95% confidence interval (CI) * at day 0 and follow-up for cows with recorded measures at both occasions.

Clinical Parameters	n	Day 0		Follow-Up	
		%	(95% CI)	%	(95% CI)
Lameness day 0 and day 1–2	89	75.2	(58.3–95.6)	44.9	(32.1–61.2)
Lameness day 0 and day 3–5	83	74.7	(57.3–95.8)	8.4	(3.4–17.4)
Affected general condition, day 0 and day 1–2	89	39.3	(27.4–54.7)	16.9	(9.4–27.8)
Affected general condition, day 0 and day 3–5	81	39.3	(27.4–54.7)	10.1	(4.6–19.2)

* Non-overlapping confidence intervals was counted as a statistically significant difference.

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
