# Peer review of "A Case-Series Report on The Use of a Salicylic Acid Bandage as a Non-Antibiotic Treatment for Early Detected, Non-Complicated Interdigital Phlegmon in Dairy Cows"

_animals, 2019, doi:10.3390/ani9040129_

Round 1

Reviewer 1 Report

There is one point I have to insist on before we can accept this

 paper. L243 the authors accepted my suggestion but added the phrase :

 ' the paper provides good evidence'. I am sorry, but that’s not the

 case. A case series doesn’t provide good evidence so I insist that the

authors remove this statement. Other than that I am happy with the

manuscript as it is now acknowledging the limitations of the study.

Reviewer 2 Report

Authors have improved the manuscript. I suggest accepting

This manuscript is a resubmission of an earlier submission. The following is a list of the peer review reports and author responses from that submission.

Round 1

Reviewer 1 Report

This could have been a very useful study on the use of salicylic acid for the treatment for interdigital phlegmon. Unfortunately, serious study design flaws led me to the decision to recommend the rejection of the paper. 

-Farmers from 24 different farms were responsible for diagnosis, treatment, follow up and record keeping. this in my opinion is introducing a large source of noise in the data and is also increasing the risk of bias.

-Some cows were excluded from the study because the farmers decided to treat them with antibiotics and no sample size calculation is provided

- However, the most serious flaw of this study is the lack of a control group. Admittedly, a negative/placebo control group could not have been included given the animal welfare implications. It is on the other hand unacceptable that a positive control group was not included in the study. The authors claim that salicylic acid is a viable alternative to antibiotics but I find their claims unsubstantiated since they did not compare salicylic acid to an appropriate antibiotic. Some clinical improvements were noticed but if antibiotics are a lot more effective than salicylic acid then I would be very cautious in recommending the use of salicylic acid as an alternative. Almost half of the studied cows still had a fever 5 days after the initiation of the treatment protocol. We need to know what would this proportion be in a group of animals treated with antibiotics. One needs to remember that interdigital phlegmon can be a very painful condition and therefore the use of antibiotics is justified. If this was a randomised control trial proving that salicylic acid was at least as effective as antibiotics I would then be the first to recommend its use.

The publication of this study could in my opinion be detrimental to cattle welfare and therefore I have serious reasons to recommend the rejection of the paper.

Given the importance of the topic I would suggest that the authors submit this manuscript as a case report removing any claims suggesting that salicylic acid may be an alternative treatment for IP and highlighting the need for a randomised control trial.

Author Response

Please find my reply in attached file.

/Ylva

Reviewer 2 Report

A manuscript of Persson Y et al “Salicylic acid bandage, non-antibiotic treatment of acute clinical interdigital phlegmon in dairy cows” was interesting, but it has many weaknesses. Especially in M&M part is written very inaccurate manner.

My main concern is lacking control group and very short follow up period. You did not controlled cure not at all after the treatment. What did you think?

How the farmers were educated to make diagnose and the treatment? How the differences between farmers was controlled in the study? Diagnosing IP is not always easy. Infected white line abscess, DD and complicated heel horn erosion can lead to misdiagnosis by the farmer.  

Usually, early treatment with antimicrobials is efficient. You don’t discuss the effect of delay with the antimicrobial treatment with severe cases. How did you ensure animal welfare aspect with this painful disease? The farmers did not use any systemic NSAIDs if I understood right. Do you really believe it?

In this manuscript is many nationally published references. Almost all references should be published in the peer reviewed journals, please try to avoid other references. It is also desirable that references are reasonable easy to access and published in language which is same (or English) with the manuscript.

Line 109 and line 145: Is this typo or did you really use this temperature for fever? Please, correct or give a full reason for this, because it affect to your results. If you want to keep this, you should find excellent reference and justify for this exceptional limit of fever.

Minor comments

Line 14               Delete comma, use hoof instead of hoofs

Line 39               Delete etc

Line 40               I find a word free ranged too prejudiced. Change the wording to loose house for                            example.

Line 48               Use microbiota instead of normal flora

Line 55               You have an English version of your guidelines available in an internet, please use it as a reference

Line 57               Add Swedish Medical Product Agency. You can also consider to remove this, because it is unnecessary for this study and there is no scientific evidence in this reference about the efficacy of this treatment.    

Line 71               Add the reference for the keratolytic properties of salicylic acid

Line 73               Add the reference for the salicylic acid properties

Line 76               Add “some” bacterial virulence properties, you have no evidence that all virulence mechanisms are inhibited.

Line 84               What was the definition of diagnose? How the farmer distinguish IP from other hoof diseases? How did you ensure that cases are uncomplicated acute IP?

Line 85               How the lameness was scored? And general condition?

Line 88               How the farmer clean the interdigital space?

Line 91               What kind of bandage was advised to use? Was there differences between herds?

Line 155             What score is not lame?

Line 169 and 191 By who? Spell it out like “Schultz and Capion (13) has earlier….”Line 170             106 cows instead of 107, why? I my mind these cows are treatment failures, which are important to identify and report.

Line 201             Correct reference.

Table 1. Trimethoprim MIC over 16, was it tested or not? 16 isolates are in the grey field, please clarify.

Table 2 and 4. Add units to the table.

Author Response

Please find a reply in attached file.

/Ylva

Round 2

Reviewer 1 Report

The authors have improved their manuscript and have provided with some justification of what they have done. I still find the lack of a control group problematic and don’t agree with not calling this work a case report or even more appropriately a case series report. A number of animals were all treated with the same treatment and followed up for a short period of time. This is the definition of a case series report and this should be made clear in the title and elsewhere.

Please consider the following title:

A case-series report on the use of salicylic acid bandage, a non - antibiotic treatment of early observed , non – complicated interdigital phlegmon in dairy cows.

Taken from the revised manuscript:

Salicylic acid bandage therefore showed to be an attractive alternative to antibiotics in the treatment of early observed , non - complicated IP

The authors do not show that salicylic acid is an attractive alternative to antibiotics since they never compared salicylic acid to antibiotics. Please remove this statement.

End of introduction:

Change the aim of the study to:

The aim of this case-series report is to…

Discussion

Please amend the following statement

Despite the mentioned weaknesses, we believe that bandage with salicylic acid is a good alternative to antibiotics in the treatment of non - complicated cases of early observed IP in dairy cows .

I suggest:

Despite the aforementioned weaknesses, we believe that a bandage with salicylic acid may be an alternative to antibiotics in the treatment of non-complicated cases of IP in dairy cows. However, a randomised positive controlled, blinded study would provide the necessary evidence that would allow us to firmly recommend this treatment instead of systemic antibiotics which is currently the only treatment of IP that has been evaluated in randomised controlled clinical trials.

You have a similar statement in the end of this paragraph but I strongly believe that this needs to be further emphasised.

Disclaimer! I am an academic working for a European Vet School. I have no affiliation whatsoever with any pharmaceutical company selling antibiotics and absolutely zero interest in the continuation of the use of systemic antibiotics for the treatment of IP. I would welcome any proven alternative. But at the same time, I strongly believe that we need to accurately report findings of studies and draw the appropriate conclusions from them. You present some promising results but the lack of a positive control mean that you cannot compare your treatment to abs and cannot suggest that it is a good alternative. But it will be useful to report your findings with the hope that they will inspire another study where salicylic acid will be compared to abs.

Please make the same changes in your conclusion (may be instead is an attractive alternative)

Author Response

Reviewer 1

Dear reviewer, Thanks for your suggestion. We fully agree, and we change the manuscript accordingly.

The authors have improved their manuscript and have provided with some justification of what they have done. I still find the lack of a control group problematic and don’t agree with not calling this work a case report or even more appropriately a case series report. A number of animals were all treated with the same treatment and followed up for a short period of time. This is the definition of a case series report and this should be made clear in the title and elsewhere.

Please consider the following title:

A case-series report on the use of salicylic acid bandage, a non - antibiotic treatment of early observed , non – complicated interdigital phlegmon in dairy cows.

A: Thanks for the suggestion. We have changed the title.

Taken from the revised manuscript:

Salicylic acid bandage therefore showed to be an attractive alternative to antibiotics in the treatment of early observed , non - complicated IP The authors do not show that salicylic acid is an attractive alternative to antibiotics since they never compared salicylic acid to antibiotics.

Please remove this statement.

A: The statement is changed.

End of introduction:

Change the aim of the study to:

The aim of this case-series report is to…

A: Changed

Discussion Please amend the following statement Despite the mentioned weaknesses, we believe that bandage with salicylic acid is a good alternative to antibiotics in the treatment of non - complicated cases of early observed IP in dairy cows .

I suggest:

Despite the aforementioned weaknesses, we believe that a bandage with salicylic acid may be an alternative to antibiotics in the treatment of non-complicated cases of IP in dairy cows. However, a randomised positive controlled, blinded study would provide the necessary evidence that would allow us to firmly recommend this treatment instead of systemic antibiotics which is currently the only treatment of IP that has been evaluated in randomised controlled clinical trials.

You have a similar statement in the end of this paragraph but I strongly believe that this needs to be further emphasised.

A: Your suggestion has been included instead.

Disclaimer! I am an academic working for a European Vet School. I have no affiliation whatsoever with any pharmaceutical company selling antibiotics and absolutely zero interest in the continuation of the use of systemic antibiotics for the treatment of IP. I would welcome any proven alternative. But at the same time, I strongly believe that we need to accurately report findings of studies and draw the appropriate conclusions from them. You present some promising results but the lack of a positive control mean that you cannot compare your treatment to abs and cannot suggest that it is a good alternative. But it will be useful to report your findings with the hope that they will inspire another study where salicylic acid will be compared to abs.

Please make the same changes in your conclusion (may be instead is an attractive alternative)"

Reviewer 2 Report

Authors have improved the manuscript, but still some issues are open. Additionally, it is very irritating that corrections are not done to the text as told or you refer to Sprecher lameness score, but actually you don’t use it. If this kind of mistakes will be in the next version I will suggest rejecting this manuscript.

You have not addressed any comment to very short follow up period. Usually, follow up period with the treatment is 2-4 weeks and very often also long term cure is taken account in scientific work. You recorded signs just during the treatment and not at all after the treatment. Imagine mastitis treatment trial where evaluation of the cure is the same day than last injection/intramammary tube.

My previous question: Usually, early treatment with antimicrobials is efficient. You don’t discuss the effect of delay with the antimicrobial treatment with severe cases. How did you ensure animal welfare aspect with this painful disease? The farmers did not use any systemic NSAIDs if I understood right.  AU: I do not fully understand this comment. Treatment with salicylic acid was probably earlier than in the cases when a veterinarian needs to be called for (mandatory in Sweden). NSAIDs are not possible to get hold of without veterinary prescription in Sweden. We have no indications that illegal use of NSAIDs is common in Sweden, why we do not think farmers used NSAIDs in this study. We hope it is also clarified in the manuscript.

In many countries antimicrobials and NSAIDs are more easily available. And in many countries veterinarians can come quite quickly to treat the cow. Farmers are not always prompt treating cow in the trimming chute. So I’m not convinced that farmers are quicker than veterinarian. The delay with the treatment appears with severe cases when the farmer first make the bandage and after two days call the vet to give antimicrobials.

You bypass in the discussion about NSAIDs which have an important function in infection diseases. Not only controlling the pain (which is important) but also decreasing inflammation.  In your country no vet-no NSAID, which is worrying with this bandage treatment.

Specific comments

Line 76 Add “some” bacterial virulence properties, you have no evidence that all virulence mechanisms are inhibited. AU: Corrected

This correction was not done.

Previous version line 85 How the general condition was scored? AU: Clarified in the MoM section.

I find this definition “General condition was scored as affected (1) or not (0).” (line 103) quite inaccurate. How did you guide the farmer to describe this? I hope you did it somehow. Please, be more precise.

You now refer lameness scoring according to Sprecher. Use it and correct the manuscript accordingly.

Line 190 “A decrease in lameness score is the most obvious symptom of an efficient treatment while fever and swelling could persist longer.” Where is the reference to this strange opinion?

Author Response

Dear reviewer,

Thanks for all your valuable comments. Please find my response in attached file.
